# Perinatal Mother-to-Child Chikungunya Virus Infection: Screening of Cognitive and Learning Difficulties in a Follow-Up Study of the Chimere Cohort on Reunion Island

**DOI:** 10.3390/v17050704

**Published:** 2025-05-14

**Authors:** Raphaëlle Sarton, Magali Carbonnier, Stéphanie Robin, Duksha Ramful, Sylvain Sampériz, Pascale Gauthier, Marc Bintner, Brahim Boumahni, Patrick Gérardin

**Affiliations:** 1Department of Pediatrics, Centre Hospitalier Universitaire (CHU) de La Réunion, 97410 Saint Pierre, Reunion, France; 2Centre d’Action Médico-Sociale Précoce Isautier, Fondation Père Favron, 97450 Saint Louis, Reunion, France; m.carbonnier@favron.org; 3Department of Pediatrics, Centre Hospitalier Universitaire (CHU) de La Réunion, 97400 Saint Denis, Reunion, France; stephanie.robin@chu-reunion.fr; 4Centre Ressources TSAF (Troubles du Spectre de l’Alcoolisation Fœtale), Centre Hospitalier Universitaire (CHU) de La Réunion, 97400 Saint Denis, Reunion, France; 5Neonatal Intensive Care Unit, Centre Hospitalier Universitaire (CHU) de La Réunion, 97400 Saint Denis, Reunion, France; duksha.ramful@chu-reunion.fr (D.R.); sylvain.samperiz@chu-reunion.fr (S.S.); 6Neuroradiology Department, Centre Hospitalier Universitaire (CHU) de La Réunion, 97410 Saint Pierre, Reunion, France; pascale.gauthier@chu-reunion.fr (P.G.); marc.bintner@chu-reunion.fr (M.B.); 7Neonatalogy Unit, Centre Hospitalier Universitaire (CHU) de La Réunion, 97410 Saint Pierre, Reunion, France; brahim.boumahni@chu-reunion.fr; 8Centre for Clinical Investigation Clinical Epidemiology (INSERM CIC 1410), Centre Hospitalier Universitaire (CHU) de La Réunion, 97410 Saint Pierre, Reunion, France

**Keywords:** arbovirus, Chikungunya virus, encephalitis, encephalopathy, central nervous system, neonate, congenital infection, cognitive function, learning difficulties, cohort study

## Abstract

In this cohort study, we evaluated the cognitive and learning difficulties of school-age children perinatally infected with Chikungunya virus (CHIKV) on Reunion Island using the *Evaluation of Cognitive Functions and Learning in Children* (EDA) battery screening test compared to the healthy children cohort used for EDA development. Of the 19 infected children, 11 (57.9%) exhibited subnormal or abnormal scores, of whom 3 were classified as high risk, and 8 were classified as at risk for cognitive and learning difficulties. Children who had encephalopathy were at higher risk for displaying at least one difficulty than non-encephalopathic children (relative risk 2.13; 95% CI 1.05–4.33). The difficulties observed affected verbal functions, non-verbal functions, and learning abilities, such as phonology, lexical evocation and comprehension, graphism, selective visual attention, planning, visual–spatial reasoning, dictation and mathematics, as well as core executive functions, such as inhibitory control, shifting, and working memory. Neurocognitive dysfunctions could be linked to severe brain damage, as evidenced by severe white matter reduction mainly in the frontal lobes and *corpus callosum* and potentially in all functional networks involved in difficulties. These results should motivate further investigation of intellectual and adaptive functioning to diagnose intellectual deficiency and severe maladaptive behaviour in children perinatally infected with Chikungunya virus.

## 1. Introduction

Chikungunya virus (CHIKV) is a positive single-stranded RNA virus belonging to the *Togaviridae* family and the alphavirus genus, transmitted by female *Aedes* mosquitoes [1]. CHIKV was first identified in Tanganyika, 1952–1953, during a polyarthritis epidemic and primarily linked to the Semliki Forest antigenic complex, half composed of Old World arthritogenic alphaviruses [2]. Though it had the initial reputation of a self-limited disease, severe neurological cases and haemorrhagic fevers, sometimes fatal, had been noticed, especially in children during the Calcutta and Vellore epidemics of 1963–1964 [3,4]. Since its 2004 re-emergence in the Indian Ocean during an unprecedented drought in coastal Kenya and an adaptive mutation in the E1 envelop glycoprotein of its East Central Southern African genotype, CHIKV has been expanding consistently on a global scale, causing large outbreaks and millions of cases worldwide [1,5]. During the first ever described high-magnitude CHIKV outbreak that occurred on Reunion Island in 2005–2006 (~300,000 infections over 17 months) [6], the conjunction of a high attack rate of a highly virulent circulating clade of the virus, the Indian Ocean Lineage (IOL), into a naïve population, along with high standard care facilities allowed the recognition of rare previously unknown severe atypical presentations of the disease, including myopericarditis, encephalitis, and Guillain–Barré syndrome [7,8,9,10,11].

In this setting, paediatricians and obstetricians described, for the first time, the perinatal mother-to-child transmission of the virus, which occurs when the future mother is highly viraemic during childbirth [12,13,14,15]. Overall, 38 neonates (median gestational age: 38 weeks, range 35–41; 22 boys, 16 girls) were infected vertically on the island, of whom 33 presented feeding problems due to suckling or swallowing disorders that further required transient enteral or parenteral nutrition [7]. Of them, 30 fulfilled the characteristic clinical picture of neonatal Chikungunya febrile prostration, which combines fever with feeding difficulties and various degrees of pain (EDIN score > 3 and paracetamol in all, nalbuphine or morphine in up to 26 neonates). Of them, 24 were lethargic or hypotonic, and for these indications, underwent cerebrospinal fluid (CSF) sampling (22 of 24 RT-PCR positive for the CHIKV genome). Of them, 14 matched the definition of a severe neonatal encephalopathy, adding seizures, status epilepticus or coma (n = 6), and/or haemodynamic disorders (n = 8) to a pathological MRI scan that showed scattered white matter, brain swelling, and/or scattered cerebral bleeds [7]. In addition, six neonates presented a haemorrhagic syndrome (five in the dominant neurological cluster) related to severe thrombocytopenia, six neonates matched the definition of myopericarditis (four in the neurological cluster), with heart failure and hemodynamic disorders (n = 2), repolarization disorders, supraventricular tachycardia, left and/or right ventricular hypertrophy, pericardium effusion, coronary dilatations, or abnormal enzymes. Taken together, these three clusters of severe neonatal disease assembled 45% (n = 15) of perinatally infected neonates and required the use of mechanical ventilation for status epilepticus or multiple organ failure (n = 9), platelet (n = 9), or fresh frozen plasma infusions (n = 7) for severe bleeding/severe thrombocytopenia.

On follow-up, 33 of these 38 perinatally infected neonates were assessed in the CHIMERE cohort study, on average, at 21 months of age, for movement/posture, coordination, language, and sociability skills using the Revised Brunet-Lézine battery and compared with 135 non-matched uninfected peers [16]. Overall, half of the infected neonates (n = 17) presented a neurocognitive dysfunction corresponding to a developmental quotient (DQ) less than or equal to 85 (−1 standard deviation), which was deemed severe with a DQ less than 70 (−2 SD) in four children. This adverse neurodevelopmental outcome affected primarily coordination and language tasks and, to a lesser extent, sociability and movement/posture. In this study, perinatal mother-to-child CHIKV infection was also an independent predictor of neurocognitive deficit once we controlled for the maternal social situation, small-for-gestational age, and head circumference, or, in another model, gestational age and breastfeeding, which, along with the abovementioned clinical description of maternal–neonatal Chikungunya, gave further credence to CHIKV as a neuroviral pathogen deserving long-term follow-up in children.

Given these observations, we hypothesized that there might be some long-lasting persistent sensorineural disorders and neurocognitive deficits in CHIKV-perinatally-infected children that could predispose them to lifelong handicaps with severe intellectual disability, mental health problems, and/or inextricable social issues.

In accordance with this hypothesis, we set up rapid ad hoc assessments of sensorineural history and neurocognitive functions of CHIKV-perinatally-infected school-age children. On this occasion, we retrieved previous evaluations of executive functions that were performed previously in *the Centres d’Action Médico-Sociale Précoces* (CAMSP), which are interdisciplinary screening centres of sensorineural and neurodevelopmental issues in France for preschool children. We also reviewed previous neuro-imagery findings.

## 2. Materials and Methods

### 2.1. Study Design and Population

We conducted a prospective follow-up study in the CHIMERE cohort in the North (CHFG: *Centre Hospitalier Félix Guyon*) and South (GHSR: *Groupe Hospitalier Sud Réunion*) university hospitals of La Réunion Island between 2 September and 15 November 2015.

The children born through the epidemic (April 2005 to August 2006) with a virologically (whole blood or cerebrospinal fluid RT-PCR) or a serologically confirmed (CHIKV-specific IgM MAC-ELISA antibodies) maternal–neonatal infection due to the vertical transmission of CHIKV during childbirth were invited to participate. The children born outside the epidemic period or diagnosed with a competing condition known to delay the acquisition of neurodevelopment milestones were not eligible for the study. The excluding conditions were as follows: very preterm birth, less than 33 weeks of gestational age, foetal alcohol spectrum disorder (FASD), autism spectrum disorder (ASD), infantile psychosis, and intellectual disability or epilepsy of other causes than neurochikungunya.

### 2.2. Exposure and Outcome Measures

Exposure measures included the best profession in the couple, as classified by the Insee (*Institut national de statistiques et d’études économiques*), ranked into four incremental groups, child characteristics at birth (gender, gestational age, birthweight, head circumference, low birthweight, and small for gestational age, 1 min Apgar score), child anthropometric characteristics at assessment (age, height, weight, head circumference), health problems (language problems, visual and hearing disorders, arthralgias, hair and nail anomalies), need for special care (psychometrician, speech therapist), and schooling (grade at primary schools, grade repetitions).

The primary outcome measure was the multi-composite EDA (*Evaluation of Cognitive Functions and Learning in Children*) test [17]. The EDA test is a user-friendly ready-to-use screening battery of neurocognitive dysfunctions that explores verbal functions (lexical evocations, lexical and syntax comprehension, plus verbal fluency as an option), non-verbal functions (token series, graphing, planning, visual–spatial reasoning and selective visual attention, plus inhibitory control, visual discrimination, gestural and constructive praxis as options), and different learning domains (reading, dictation, and mathematics) in children aged 4 to 11 years.

The EDA was developed and calibrated on 626 healthy French children taken at random from all socio-environmental backgrounds and divided into six grade levels [14]. The gender distribution in the training sample was identical in each stratum, with a male-to-female ratio of 1.01. The distribution of parental socio-professional categories was homogeneous in the six different grade levels.

EDA scales classify cognitive functioning and children as normal with low risk when scores are equal or higher than −1 standard deviation (SD), subnormal and at risk of difficulties when they fall between −1 SD and −2 SD, and abnormal with high-risk difficulties when they are lower than −2 SD.

The EDA test has been used routinely in France for a decade as the tool of choice in the screening for speech (e.g., for indicating the need for a speech therapist) [18] and learning disorders (e.g., for proposing a specialized support network) [19], but it has been seldom used for research beyond a single regional cohort of very preterm (VPT) children [20].

Secondary endpoints included former assessments of executive functions and related behavioural processes retrieved from the *Behaviour Rating Inventory of Executive Function* preschool version (BRIEF-P) [21] and various tests superimposing BRIEF-P scales for the evaluation of inhibition, shifting, emotional control, initiation, working memory, and planning/organization. These included the preschool version of the Trail Making Test (TMT) [22], Dimensional Change Card Sort (DCCS) test [23], NEPSY-derived statue test [24], hands game, yes/no game, tower builds, and various span tasks. These tests were passed by a neuropsychologist for seven of the ten children referred to CAMSP centres in the setting of screening neurodevelopmental disorders for children aged 0 to 6 years. BRIEF indexes have been standardized on age and sex to a mean of 50 and an SD of 10 in a population of American children, with scores higher than +1 SD defining difficulties [25]. The other tests have been used similarly, with +1 SD beyond the observed mean indicating abnormality.

### 2.3. Statistical Analysis

The characteristics of CHIKV-perinatally-infected children are described as percentages, medians, and interquartile ranges (Q_1_–Q_3_). EDA subscales are reported as T-score means with their standard deviations and 95% confidence intervals (95%CI) or Z-score means with extreme values. The EDA result for the classification of CHIKV-perinatally-infected children into normal/low risk, subnormal/at risk, or abnormal/high risk is reported for each subscale using a vertical bar diagram distinguishing the proportions of children in each risk category. The relative risks of cognitive and learning difficulties (y1: child with high risk or at risk, y0: child with low risk as reference) are expressed for encephalopathic children versus non-encephalopathic children (reference).

Means of EDA T-scores were compared between CHIKV-perinatally-infected and healthy children from the French national EDA validation cohort using Student’s *t*-test with Welch corrections to account for both group size asymmetries and differences in variances and between CHIKV-infected encephalopathic and non-encephalopathic children using the Mann–Whitney U test. Means of EDA Z-scores of CHIKV-perinatally-infected children were compared across strata of maternal education levels using the Kruskal–Wallis H test. Differences in mean Z-scores between CHIKV-infected encephalopathic and non-encephalopathic children, between CHIKV perinatally infected and VPT children of the Lorraine regional cohort [20], are also reported in vertical bar diagrams.

Statistical analyses were performed with Stata^®^ (v16·1, StataCorp, College Station, TX, USA, 2019). For all analyses, observations with missing data were ruled out, and a two-tailed *p*-value less than 0.05 was considered significant.

## 3. Results

### 3.1. Population Characteristics

Of the 33 infected children enrolled in the CHIMERE cohort between 15 April 2006 and 2 August 2006, 3 were lost-to-follow-up, and 9 refused to participate, leaving 21 children willing to contribute to the new evaluation. Of these 21 infected children, 12 had been classified as non-encephalopathic (or non-severe prostrated), and 9 were classified as encephalopathic at the time of the CHIMERE study (at 21 months of age, on average; range 15.8 to 26.7 months). This represented 57% of the non-severe and 75% of the severe infected children, among whom four out of the five children with postnatal onset microcephaly and two out of the four children with cerebral palsy in 2007–2008 (Figure 1).

Participating children were aged 9.8 years, on average (median age 9.6 years, extremes 9.5–10.4; cumulative follow-up of 205.3 person-years) and composed of 12 boys and 9 girls. Three of the four microcephalic children still had their head circumference below the third percentile (<−2 SD), and for the fourth, head growth had catch-up on a trajectory within the −2 and −1 SD small head growth corridor. Five children (26.3%) reported persistent arthralgia, which consisted of undifferentiated oligoarthralgia. Thirteen children (61.9%) had experienced language disorders requiring the help of a speech therapist, and eleven (52.4%) had encountered vision problems (cortical blindness in 1, strabismus in 4, refraction disorders in 7, type of myopia in 3, hypermetropia in 2, and astigmatism in 2). Five encephalopathic children required special education. Five infected children had repeated a grade in primary school (3 encephalopathic and 2 non-encephalopathic).

The characteristics of children perinatally infected with CHIKV are presented in Table 1.

### 3.2. Primary Outcome Measure

#### 3.2.1. EDA Result in CHIKV-Perinatally-Infected Versus Healthy Children

Of the 21 children perinatally infected with CHIKV on Reunion Island, 19 could be assessed using the EDA test, and two could not due to overwhelming handicaps. These 19 infected children displayed lower scores in verbal functions, such as phonology (*p* = 0.0338), lexical evocation (*p* = 0.0301), and lexical comprehension (*p* = 0.0191) than the 626 French healthy children, as well as non-significant trend towards low performances in both syntactic expression and comprehension (Table 2). Consistently, they performed worse in tasks exploring non-verbal functions, such as graphism (*p* = 0.0001), selective visual attention (*p* = 0.0039), planning (*p* = 0.0238), and visual–spatial reasoning (*p* < 0.0001). With respect to learning, CHIKV-perinatally-infected children exhibited lower averages in dictation (*p* = 0.0168) and mathematics (*p* = 0.0061) but not in reading exercises.

The EDA battery identified a high risk for cognitive and learning difficulties in three children with abnormal scores, classified eight children with at least one subnormal performance as at risk, and eight children with normal performances as low risk. This means that 57.9% (11/19) of CHIKV-perinatally-infected children performed lower than children of similar age, sex, and socio-environmental background (in the case where the 2 different cultural backgrounds would have no significant influence on scores). The risk classification of the 19 CHIKV-perinatally-infected children is displayed in Appendix A, Figure 2A.

#### 3.2.2. EDA in Chikungunya Encephalopathic Versus Non-Encephalopathic Children

The seven encephalopathic children who could be assessed exhibited lower scores than the twelve non-encephalopathic children in phonology (*p* = 0.0407), lexical evocation (*p* = 0.0049), and in both syntactic expression (*p* = 0.0023) and comprehension (*p* = 0.0004). They also performed worst in graphism (*p* = 0.0184), selective visual attention (*p* = 0.0257), and planning (*p* = 0.0329) tasks. Finally, encephalopathic children presented a non-significant trend towards lower performances in the three learning abilities (Appendix A).

The encephalopathic children who were tested tended to have a higher risk of at least one difficulty than non-encephalopathic children (85.7% vs. 41.7%, *p* = 0.1473; relative risk 2.06, 95%CI 0.98–4.29). This trend was significant if we assumed that the tasks that could not be completed were due to overwhelming hurdles hampering testing (88.9% vs. 41.7%, *p* = 0.0375; relative risk 2.13, 95%CI 1.05–4.33), which was indeed the instance for the two encephalopathic children with *sequelae* (i.e., cerebral palsy) who were ruled out from tests.

#### 3.2.3. EDA in Chikungunya Non-Encephalopathic Versus Healthy Children

The twelve non-encephalopathic children displayed lower scores than the 626 French healthy children taken as controls in the three non-verbal functions: graphism (*p* = 0.0012), selective visual attention (*p* = 0.0165), and visual–spatial reasoning (*p* < 0.0001), as well as in mathematics (*p* < 0.0001). By contrast, they performed better than controls in syntactic expression (*p* = 0.0131) (Appendix A).

#### 3.2.4. EDA in Chikungunya Non-Encephalopathic vs. Non-Disabled Very Preterm Children

The twelve non-encephalopathic children tended to have higher Z-scores than those found in non-disabled VPT children, except for mathematics, although it could not be tested statistically due to missing information in the control group (Appendix A).

#### 3.2.5. EDA in Chikungunya Encephalopathic vs. Disabled Very Preterm Children

The seven encephalopathic children tended to have slightly lower Z-scores than those found in disabled VPT children, except in mathematics, for which the difference was sharp (>3 SD), although it could not be tested statistically in the control group (Appendix A).

#### 3.2.6. EDA in CHIKV-Perinatally-Infected Children According to Maternal Education

Maternal education level had no detectable influence on CHIKV-perinatally-infected children’s skills, as shown by similar Z-scores for children from mothers at elementary school, high school, bachelor’s degree, or college, as best schooling level (Appendix A).

### 3.3. Secondary Outcome Measures

Of the seven children referred to CAMSP centres before six years of age, five had scored higher than +1 SD on at least one BRIEF-P scale and were detected with potential dysexecutive disorders (4 encephalopathic, 1 non-encephalopathic). In further detail, these children were described as “distractible”, “impulsive”, and/or “mentally stiff”. Of them, three children (1 non-encephalopathic) had exhibited abnormal inhibition control, which was further confirmed by the impossibility of connecting a series of dots and completing TMT part A within 90 s, part B within 300 s, as well as abnormal responses, sometimes with motor persistence in hand games, yes/no game, tower builds, and, to a lesser extent, in the statue test (2/3). Two encephalopathic children had been detected with difficulties in shifting between activities, which was confirmed with a pattern of mental inflexibility (perseverations) during the post-switch phase of the DCCS test. Emotional control was normal on examination despite parental reports of emotional lability in two encephalopathic children. Working memory had been detected as abnormal in three children (1 non-encephalopathic), which was further confirmed in updating difficulties during diverse span tasks. Planning had been detected as compromised in four children (1 non-encephalopathic child), which was confirmed with TMT and London tower tests.

The putative neuro-anatomical–clinical correlates between the observed cognitive dysfunctions and neuro-imagery findings are displayed in Appendix A.

## 4. Discussion

### 4.1. Key Findings

In this study, we confirm the impact of perinatal mother-to-child Chikungunya virus infection on school-age child neurodevelopment previously assessed around the age of two in the CHIMERE cohort [16]. The study only involved half of the children infected during the epidemic and presents several caveats: the study was not controlled other than by a theoretical target population of metropolitan children of comparable age and sex; the choice of the EDA screening tool as a primary outcome measure did not allow definitive conclusions on child’s individual neurodevelopment; our results provide important information for the triage assessment of CHIKV-perinatally-infected children, the search for intellectual deficiency (i.e., mental retardation), their management, and the relief, control, and prevention of related handicaps.

First, although the new cohort was not fully representative of the CHIMERE cohort, the prevalence of 57.9% of abnormal/subnormal EDA tests at around age ten in infected children is remarkably consistent with the 51% of abnormal/subnormal Brunet-Lézine tests described around the age of two (defined as global neurodevelopmental delays) [16], which likely indicates that the neurodevelopment of these children has not been the subject of effective, sustained interventions, or, for the most seriously affected ones, has escaped the potential benefits of neuroprotection, yet hoped given the expected cerebral plasticity.

Second, the extent of neurocognitive dysfunctions correlates with the severity of neonatal infection, this being larger in encephalopathic than in non-encephalopathic children, which also corroborates previous findings from the CHIMERE cohort [16]. Thus, for the former, dysfunction was overwhelming, affecting both verbal and non-verbal skills, as well as executive functions (working memory, shifting, inhibition) and related mental processes (emotional control, initiation, and planning). For the latter, dysfunction was selective and primarily affected non-verbal skills (with exception of planning). However, it could seem haphazardous from this analysis to determine whether dysfunctions in visual reasoning and mathematic abilities were due to the infection, a poor fit of the EDA test for Reunionese Creole-speaking children, or both. Additionally, it is unknown whether differences between the infected population and French healthy controls were due to the inadaptation of EDA to the Creole context. Interestingly, CHIKV-perinatally-infected encephalopathic children seemed to display lower EDA Z-scores than a cohort of disabled very preterm children observed in metropolitan France (with the exception of visual attentive selection), while the non-encephalopathic children scored better than non-disabled very preterm children (with the exception of mathematics) from the same Lorraine cohort.

Of note, EDA scores were not driven by maternal education level, which gives further credence to CHIKV neuropathogenesis as the cause of neurocognitive dysfunction.

### 4.2. Strengths and Limitations

First, the new cohort was nested in the CHIMERE cohort, which warrants the certainty of disease classification into encephalopathic and non-encephalopathic children, excluding other causes of encephalopathy [16].

Second, we checked early childhood histories to eliminate competing conditions that affect the neurodevelopmental skills of children, which, together with the previous observation of early neurodevelopmental delays in the cohort, the negligible role of maternal education on performance in the study population, and the growing body of evidence for CHIKV triggering neuropathogenesis (reviewed in [25,26,27]), supports the pathogenic role of the virus in the observed neurocognitive dysfunction.

This work also has several limitations.

First, the study did not include a dedicated, purposely designed control population to ensure the validity of the EDA development population of healthy French children as a group to be compared with CHIKV-perinatally-infected children. This might have slightly overestimated the difference in EDA scores between the infected group and the control group. However, the comparisons with very preterm groups fulfilled the expected direction, which makes this overestimation unlikely to bias the overall sense of our findings.

Second, we were unable to collect information about nine infected children whose parents refused for the child to participate, which might have introduced a referral bias, a selection bias that makes participants frequently different from non-participants. However, our group of interest represented 57% of the non-severe and 75% of the severe infected children, which may ensure a relative representativeness of CHIKV-perinatally-infected children.

Third, neurodevelopment skills were assessed using the EDA battery, a user-friendly, ready-to-use neurodevelopment screening tool that can be handled with minimal training by non-skilled professionals, including paediatricians, during consultations. Moreover, the EDA test was administered in unblinded conditions to the disease group, which might have introduced a differential classification bias towards a more terrible assessment of the encephalopathic group. In this instance, two children were known to live in very poor socio-environmental conditions, which might have introduced an evaluation bias, the examinator being tempted to lower the scores of these children, given the known fact that living in deprived neighbourhoods often hampers children’s neurodevelopment through multicomposite disadvantage [28].

Fourth, we did not complete our investigation with a functional MRI study to give further credence to the etiopathogenic role of CHIKV neonatal encephalitis in children diagnosed with encephalopathy and brain lesions, which would have represented a gold standard to link neurocognitive dysfunctions to neuropathologic anatomy.

### 4.3. Interpretation

In this study, we report the consequences of perinatal mother-to-child CHIKV infections with an unparalleled 10 year’s hindsight, which allows more understanding of the brain functioning of infected children through the vertical transmission of the virus. By the age of ten, the examination of cerebral functions is more refined, enabling us to pinpoint with clinico-radiological correlates the precise location of damaged brain areas.

Each hemisphere has four external lobes: the frontal lobe, temporal lobe, parietal lobe, and occipital lobe, plus two lobes hidden in cortical folds, the limbic lobe and the insula. The brain’s functions are divided between these lobes. Each lobe can perform several functions and communicate with each other through synaptic connections [29,30]. Damage to one of these areas or to the neurocircuits between them disrupts the associated function [31].

Though we did not perform new MRI scans for the study, several associations can be drawn between neurocognitive dysfunctions and previous neuroimagery findings. These images uncover white matter lesions rather than grey matter lesions, which suggests that the lesion substratum lies in neural networks connecting grey to white matter in grey centres.

Study findings reveal that CHIKV-induced brain damages mainly affect verbal and non-verbal functions and, to a lesser extent, the executive functions and their goal-directed-related behaviours, especially in the most severely affected encephalopathic children.

Difficulties in speech processing (i.e., phonology) are left-lateralized and may relate to damages connected to the left superior temporal gyrus (STG), Broca’s area subdivisions such as the left inferior frontal gyrus (IFG) and supramarginal gyrus (SMG), as well as the dorsal part of the left inferior parietal lobule (dIPL), which is also involved in semantic processing [32]. Dysfunction led to language delays, requiring a speech therapist. Importantly, two of three encephalopathic children with altered speech processing presented an impaired initiation of inhibitory control, which depends on the right IFG [33]. Interestingly, these two children presented an atrophy of the anterior body and *genu* of the *corpus callosum* and *centra semiovale*, which underpins interconnectivity between the two frontal lobes [29,31].

Difficulties in naming and word meaning (i.e., lexical evocation and lexical comprehension) may be connected to the dorsal premotor (PMd) cortex in the left frontal lobe [34]. This association was supported in one encephalopathic child by the conjunction of a previously observed impaired fine motor coordination at the time of the CHIMERE cohort and a concomitant dysfunction in planning. This multiple dysfunction was related to the abovementioned lesions of white matter masses and significant astrogliosis with remodelling (“squaring”) of the frontal horns.

Difficulties in grammar and links between linguistic units (i.e., syntactic expression and syntactic comprehension) may relate to the left posterior inferior frontal gyrus (pIFG) and posterior middle temporal gyrus (pMTG) of the Broca’s area, respectively [35], despite the absence of other associations crediting the involvement of these brain regions. In addition to the abovementioned lesions, one of the two children investigated for this dysfunction presented a very thin splenium of the *corpus callosum*, which interconnects temporal lobes.

Difficulties in graphic thinking (i.e., graphism) may be connected to right and left prefrontal cortices (PFC), which contain dorsolateral prefrontal cortices (DlPFC), anterior cingulate cortices (ACC), and middle frontal gyri (MFG), which are known to contribute to the planning and refining of graphic tasks through idea generation and idea production (under the control of the left precentral gyrus, a subdivision of the left DlPFC) [36]. These associations were made plausible in two encephalopathic children by previously observed difficulties in emotional control (dependent on ACC), working memory, shifting (flexibility), and planning (all three dependent on DlPFC), as well as concomitant difficulties in selective visual attention and visual–spatial reasoning, with all of these functions being supported by large-scale white matter functional networks, including the dorso-fronto-parietal (D-FPN), the lateral fronto-parietal (L-FPN) and midcingulo-insular (M-CIN) networks [31].

Selective visual attention describes the tendency of visual processing to be confined to relevant stimuli dedicated to goal-directed behaviours while irrelevant information is inhibited [37]. It is a complex multimodal cognitive function involving several types of attention [35] mediated by the D-FPN (i.e., dorsal attention network), a large-scale neurocircuit that includes connections between the DlPFC, the “frontal eye fields” (FEFs), and the IFG in the frontal lobe, as well as with the superior parietal lobule, the anterior insula (AI), the posterior cingulate cortex (PCC), and the occipital cortices outside [31,38]. In this study, it was considered at risk of difficulty in three encephalopathic children.

Difficulties in planning (e.g., anticipation) may disrupt the interplay between the PFC (the premotor structure acting as “simulator”, testing possible actions) and the hippocampus (used in memory formation and storage) [39]. In this study, dysfunction in planning actions was first observed in three encephalopathic children with the BRIEF-P, but only in two of them, a few years later with the EDA, which suggests improvement in the third child or evaluation bias. Interestingly, the BRIEF-P planning score correlated with the working memory score in those three children, which both illustrate the interplay between the PFC and the *amygdala*–*hippocampus* circuit [40] and the supportive role of working memory in complex mental functions such as planning [41].

Difficulties in visual–spatial reasoning may relate to biparietal cortices and lateral bioccipital cortices [41]. They were observed in one encephalopathic child with extensive neurocognitive dysfunction, including selective visual attention, shifting (flexibility), two visually-guided functions [31], graphism, and planning. In this child, dysfunction led to constructive apraxia linked to *corpus callosum* thinning, including the *splenium* and *rostrum.*

Difficulties in reading and dictation, two upper cognitive functions, whether they were associated with difficulties in speech processing or not, may be connected to temporal and parietal lobes [42] and associated with lower myelination in bilateral *thalami*, as well as left and right anterior and left posterior limbs of the internal capsula, bilateral *centrum semiovale*, and *splenium* [43]. They were initially observed in 13 children (7 encephalopathic and 6 non-encephalopathic) and improved during childhood in almost all, but not for three encephalopathic children, for whom they remained detectable at ten using the EDA test. Interestingly, one of these children had presented T1 hypersignals of both internal capsula and a thin *splenium* on early MRI scans, which both disappeared at the 24-month control.

Difficulties in computation (i.e., mathematics), another upper cognitive function, may relate to damages to the left superior frontal gyrus (SFG), right superior parietal lobule [44], as well as to white matter microstructure (e.g., lower fibre density, thinning) [45]. They were observed in one encephalopathic child from a non-French speaking mother tongue—one encephalopathic child with splenium hypoplasia and his non-encephalopathic twin sister, born preterm. In addition, mathematic scores were lower in non-encephalopathic children, as compared to French healthy or VPT neonates, which might indicate an evaluation bias.

Executive functions are high-level cognitive processes involving abilities such as working memory (updating), set-shifting (flexibility), and inhibition. These cognitive functions engage simultaneously during complex mental processes such as abstraction, planning, reasoning, and decision-making [31]. They are pillars for adaptive functioning, which conditions resilience in daily life and future social insertion [46].

Working memory, the engine of learning [47], relates to the ability to transiently process, maintain and use information input from diverse sensorial stimuli to serve ongoing tasks and goal-directed behaviours [31]. Difficulties in updating may relate to damages to: 1. superior longitudinal fasciculus (SLF-I, SLF-II), which ensures spatial attention and spatial location of salient stimuli via the D-FPN and transfers information from the phonological store of the dIPL to Broca’s area (phonological loop); 2. the frontal aslant tract (FAT) that connects IFG, ACC, and medial regions of the SFG via the L-FPN affects the inhibitory control and speech initiation; or 3. the fronto-striatal tract (FST) that connects the SFG to the caudate nucleus and impairs motor inhibition, speech initiation, and flexible manipulation of verbal information. Interestingly, the three children with dysfunction had problems with inhibitory control (and planning), two with speech initiation, and one with visual attention, and all had difficulties performing learning tasks.

A changing environment requires a flexible reorientation of attention from one stimulus or mental representation to another to adapt behaviour to new situations or to solve new problems [31]. Difficulties in shifting may result from damages to SLF (SLF-III, SLF-II), the *arcuate fasciculus* and, to a lesser extent, the middle longitudinal *fasciculus* (MLF) and callosal fibres (*tapetum*), which together form the L-FPN, a multi-demand cognitive control network spreading bilaterally between the inferior frontal *sulcus* (IFS), AI, and adjacent frontal *operculum* (AI/FO), pre-supplementary motor area, and adjacent ACC, in an around intraparietal *sulcus* (IPS) [48]. Interestingly, one of the two encephalopathic children with dysfunction exhibited deficits in working memory and inhibitory control at preschool age, which may relate to the L-FPN and D-FPN, which have common executive abilities [31].

Inhibition is a multi-faceted concept that encompasses behavioural and cognitive processes to deliberately suppress dominant, automatic responses and resist interference from environmental cues to focus on information that is relevant to the accomplishment of ongoing tasks [31]. Difficulties in inhibition response may relate to disconnections among SLF (SLF-III, SLF-II), white matter in the IFG, MFG, AI, and the right anterior thalamus. Interestingly, the three children with inhibitory control dysfunction had deficits in working memory, and one had difficulties in set-shifting, which suggests inhibitory control may be important to prime or maintain working memory in new tasks and goal-directed behaviours [49].

### 4.4. Generazibility

The CHIMERE (Chikungunya Mère-Enfant) cohort on Reunion Island is unique in the world. The opportunity to study this cohort was created by the conjunction in a short time and space of a high magnitude CHIKV outbreak into a naïve population with a highly virulent clade of the virus (IOL), which is believed to have increased the ability of CHIKV to bypass the protection of natural barriers, including the placental and blood–brain barriers [50,51].

Given the abovementioned and the fact that CHIKV does not behave like a TORCH pathogen [25], our findings may not be replicated or generalizable to other pathogens and study contexts. We believe, however, that some clinical comparisons should be made with the neurodevelopmental outcomes and neuroimagery findings of three conditions known to impair child mental health, namely very preterm birth, FASD (including alcohol-related neurodevelopmental disorders), and congenital Zika virus (ZIKV) infection [52,53,54].

Children born very preterm (<32 weeks’s gestation) or with very low birth weight (<1500 g) present a wide range of neurocognitive dysfunctions, including dysexecutive disorders in working memory and shifting, visual–motor dyspraxia, dyslexia, and dyscalculia, which impair learning and schooling. In the Lorraine cohort, this yields 32% of VPT children with EDA scores within the normal range (>−1 SD) in all subscales (28% in children with neurodevelopmental abnormalities, 35% in children without neurodevelopmental abnormalities) [20], a result a priori more pejorative than for perinatally infected children with CHIKV in our cohort (42% overall, 14% in encephalopathic children, 58% in non-encephalopathic children, respectively).

VPT children exhibit volumetric, morphologic, and microstructural alterations in subcortical and temporal cortical regions that result in lower intracranial, total brain, grey matter, and white matter volumes, especially in the *hippocampus*, *amygdala*, basal *ganglia*, *cerebellum*, *corpus callosum*, and *cingulum*, with higher volumes in the ventricles; reduced thickness of IFG (pars triangularis and pars orbitalis), MTG, SMG, and IFP cortices; lower white matter radial diffusivity and fibre density; and higher white matter mean diffusivity [52].

Children with FASD present cognitive impairments and behaviour problems (ARND), including dysexecutive disorders and altered mental processes in shifting, verbal fluency, inhibition, conceptual reasoning, planning and organization, decision-making and emotional control that result in hyperactivity, impulsivity, poor communication and poor social skills, cue alcoholic reactivity, and to a lesser extent, dyslexia and dyscalculia.

These dysfunctions align with structural anomalies in several cortical and subcortical brain regions intimately involved in the control of risky behaviours, as well as in reinforcing stimuli. These include disproportionate reductions in frontal lobes despite increased grey matter densities in MFG, IFG, and DlPFC, *insula*, ACC, *accumbens nucleus*, lower grey matter volumes in the *hippocampus* and basal *ganglia* (especially *caudate nucleus*), lower white matter volumes in the superior and inferior longitudinal *fasciculi* (SLF, ILF), *corpus callosum* (*genu* and splenium), *corona radiata*, and *cerebellum*, displacement of the *corpus callosum* and altered grey matter asymmetry [53,54]. So far, no study on FASD children has reported EDA tests, which prevents comparison to perinatally infected children with CHIKV.

Children born with congenital Zika syndrome (CZS) have a very poor prognosis with extremely low performances in motor, cognitive, and language development due to extensive brain damage in both grey matter and white matter, and practically all feature severe forms of cerebral palsy [55]. To the best of our knowledge, no study has yet linked neurocognitive dysfunctions to specific damaged brain areas in CZS children. Given potential neuropathology mimicking CHIKV neonatal encephalitis in offspring from pregnant pigtail macaques infected with ZIKV at 33 weeks gestation, on average (white matter damage, postnatal microcephaly) [56], as observed in the CHIMERE cohort, we postulate the possibility of similar neurocognitive dysfunctions and prominently white matter lesions in the instance of late congenital ZIKV infection.

### 4.5. Implications

Our study confirms that maternal–neonatal CHIKV infection may lead to cognitive and learning difficulties at school age through the *sequelae* from severe brain damage caused by CHIKV neonatal encephalitis. These lesions, almost exclusively located in the white matter, affect both the verbal and non-verbal dimensions of neurodevelopment, as well as core executive functions, which together result in multidimensional disorders of coordination, language, and child’s perceptual and conceptual reasoning, functions mainly associated with the frontal lobe.

For individual risk prediction and clinical practice, the EDA test may be convenient for screening cognitive and learning difficulties in perinatally infected children with CHIKV with the aim to identify the children who need further testing to diagnose severe disabilities (e.g., mental retardation or intellectual deficiency, usually defined with a Full Scale Intellectual Quotient of the Wechsler Intelligence Scale for Children of 70 or less; severe maladaptive functioning, usually defined with a Vineland Adaptive Behaviour Scale score of 70 or less). These children may benefit from developmental programs targeted at function, as for VPT and FASD children, even though their effectiveness depends on the earliness of screening and their impact is still debated beyond preschool age [57,58].

In our cohort, neurocognitive dysfunctions and brain damage varied greatly from one child to another, and hopefully, half of the children detected with global neurodevelopmental delays at preschool age, using the Brunet-Lézine in the CHIMERE cohort, seemed to have catch-up at school age using the EDA test as a screening tool. For the children concerned, this might reflect the positive effects of early developmental interventions in CAMSP centres at preschool age along with brain plasticity (e.g., the malleability of postnatal brain development). The concept of brain plasticity involves increased myelination and synaptic pruning of the white matter, cortical thinning of the grey matter, and the maturation of the “connectome”, an ensemble of overlapping modules with their structural substrates aiming to supply a resilient individual functional network [59].

For public health purposes, the EDA battery or similar user-friendly ready-to-use screening tests could be incorporated into screening strategies and guide further testing when the abovementioned gold standard tests are impossible to implement.

For clinical rese arch purposes, this study helped to build up new research aimed at assessing the intellectual and adaptive functioning of CHIKV-perinatally-infected children in adolescence. This study has been conducted, and its results are submitted for publication concomitantly to this article [60].

## 5. Conclusions

The results of our study shed light on the neurocognitive dysfunctions and putative brain damage found in perinatally infected children with Chikungunya virus while proposing anatomo-clinical correlates from previous neuro-imagery findings to explain the observed dysfunctions. As previously shown in the CHIMERE cohort, dysfunctions were more severe in encephalopathic children than in non-encephalopathic children, which deserves further investigation of intellectual and adaptive functioning to diagnose mental retardation or intellectual deficiency and severe maladaptive behaviour.

## Figures and Tables

**Figure 1 viruses-17-00704-f001:**
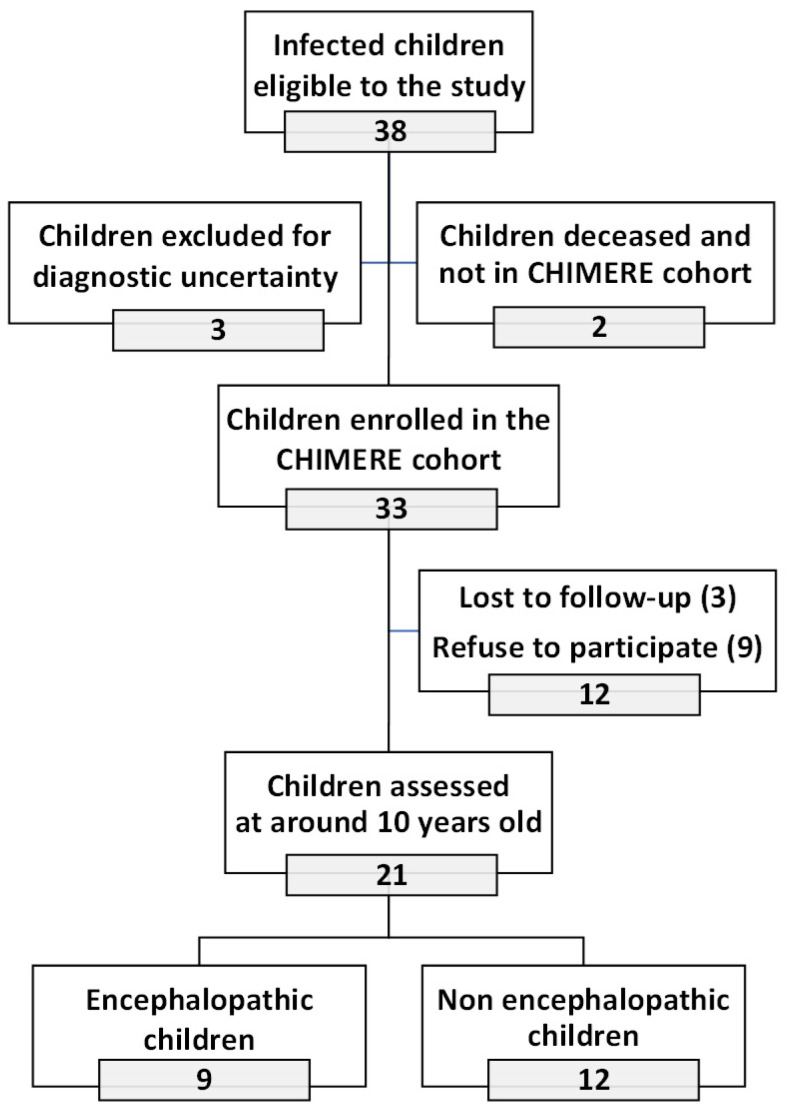
Study population.

**Figure 2 viruses-17-00704-f002:**
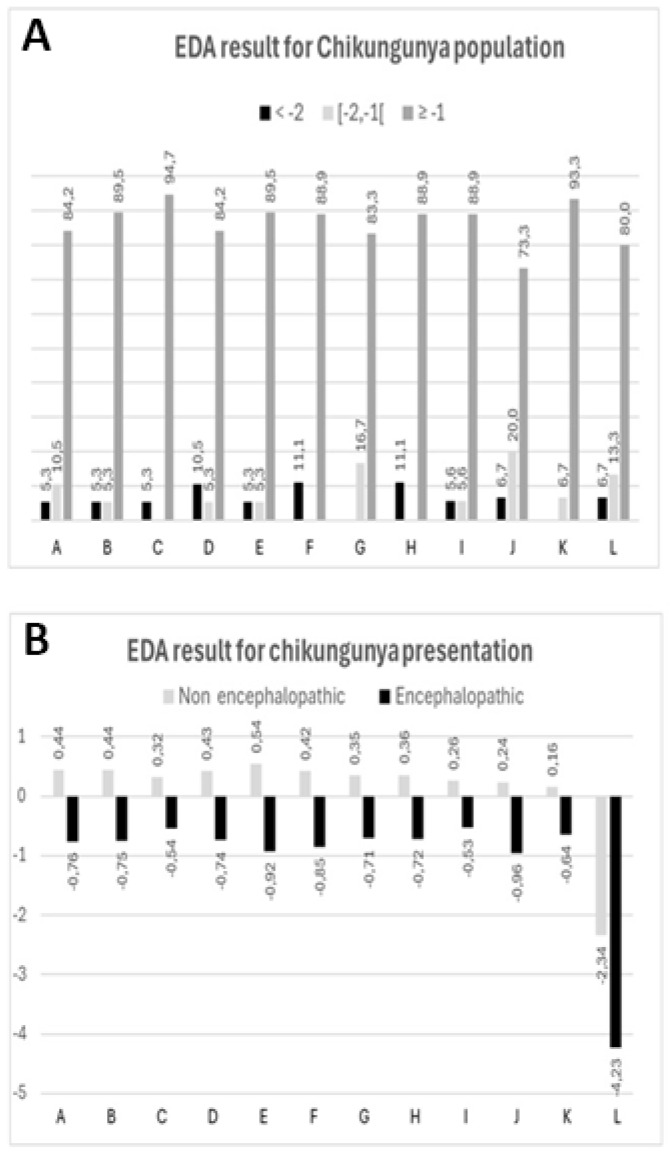
EDA Z-scores. (**A**) Percentage of CHIKV-perinatally-infected children in the three risk categories based on Z-scores. (**B**) Mean Z-scores for each EDA subscale and disease presentation (encephalopathic vs. non-encephalopathic). A: Phonology, B: lexical evocation, C: lexical comprehension, D: syntactic expression, E: syntactic comprehension, F: graphism, G: selective visual attention, H: planning, I: visual–spatial reasoning, J: reading, K: dictation, L: mathematics.

**Table 1 viruses-17-00704-t001:** Characteristics of 21 children perinatally infected with Chikungunya virus, screened at around 10 years old for neurocognitive dysfunction, Reunion Island, 2015–2016.

	n = 21	(%)
** *Parental characteristics* **		
**Higher-graded profession in the couple**		
Group 0	11	(26.8)
Group 1	20	(48.8)
Group 2	5	(12.2)
Group 3	5	(12.2)
** *Child characteristics at birth* **		
**Gender**		
Male	12	(57.1)
Female	9	(42.9)
**Gestational age**		
(weeks gestation; median, Q_1_–Q_3_)	38	(37–39)
**Birthweight**		
(Grams; median, Q_1_–Q_3_)	2770	(2330–3170)
**Low birthweight**		
No (BW ≥ 2500 Gr)	15	(71.4)
Yes (BW < 2500 Gr)	6	(28.6)
**Small for gestational age**		
No (BW > 10th centile)	18	(85.7)
Yes (BW 10th–3rd centile)	1	(4.8)
Yes (BW < 3rd centile)	2	(9.5)
**Head circumference at birth**		
(cm; median, Q_1_–Q_3_)	33.5	(32–35)
Normal sized, −1 SD to +2 SD	17	(80.9)
Small sized, −2 SD to–1 SD	3	(14.3)
Microcephaly < −2 SD	1	(4.8)
**Apgar score at 1 min** (cm; median, Q_1_–Q_3_)	10	(10–10)
≥7	19	(90.5)
<7	2	(9.5)
** *Child characteristics at 10 years* **		
**Age** (years; medians, Q_1_–Q_3_)	9.9	(9.7–10.0)
**Height**		
(cm, medians, Q_1_–Q_3_)	142	(136–145)
**Weight** (KG; median, Q_1_–Q_3_)	35.6	(30.0–42.0)
**Head circumference**		
**at follow-up** (cm; median, Q_1_–Q_3_)	54	(53–56)
Normal sized, −1 SD to +2 SD	11	(57.9)
Small sized, −2 SD to −1 SD	5	(26.3)
Microcephaly < −2 SD	3	(15.8)
**Health problems**		
Language disorders	13	(61.9)
Vision problems	11	(52.4)
Hearing disorders	0	(0)
Arthralgia	5	(23.8)
Alopecia	5	(23.8)
** *Schooling* **		
**Grade at primary school**		
Elementary school 2nd year (CE2)	1	(4.8)
Middle school 1st year (CM1)	11	(52.4)
Middle school 2nd year (CM2)	4	(19.0)
Special education class *	5	(23.8)
**Grade repeat**	5	(23.8)

Data are numbers and column percentages or medians and interquartile range (Q_1_–Q_3_) when specified. Insee (Institut national des statistiques et études économiques) classification of the 7 French socio-professional categories was graded into 4 incremental groups according to average incomes: 0: no profession (never or retired), 1: farmers, craftsmen, traders, and workers, 2: employees and intermediate professions, 3: executive and upper professions. * IME: Institut médico-éducatif, IMP: Institut médico-pédagogique, SESSAD: Service d’Education Spéciale et de Soins à domiciles.

**Table 2 viruses-17-00704-t002:** Performances on EDA (Assessment Scale of Child Cognitive and Learning Functions—First edition) scales in a cohort of 21 children perinatally infected with Chikungunya virus, Reunion Island, 2015–2016, and in a French national cohort of 626 healthy children (controls).

Exposure Group	Healthy Controls			CHIK+			
**Functions**	n = 626			n = 19			
** *Verbal (n = 19)* **	Mean	SD	(95% CI)	Mean	SD	(95% CI)	*p-value*
Phonology	19.8	0.6	19.7–19.8	16.9	5.5	14.2–19.6	**0.0338**
Lexical evocation	54.1	3.3	53.8–54.4	48.1	11.1	42.7–53.5	**0.0301**
Lexical comprehension	32.8	1.3	32.7–32.9	28.9	6.6	25.7–32.1	**0.0191**
Syntactic expression	17.6	1.5	17.4–17.7	16.2	5.9	13.3–19.0	0.3151
Syntactic comprehension	27.2	2.6	27.0–27.4	24.9	6.6	21.7–28.1	0.1468
** *Non-verbal (n = 18)* **							
Graphism	7.0	1.8	6.8–7.1	4.4	2.1	3.3–5.4	**0.0001**
Selective visual attention	23.4	4.7	23.0–23.8	18.9	5.7	15.2–20.9	**0.0039**
Planning	8.6	1.5	8.4–8.7	6.9	2.9	5.4–8.3	**0.0238**
Visual–spatial reasoning	24.9	2.5	24.7–25.1	7.9	3.3	6.2–9.5	**<0.0001**
**Learnings (n= 15)**							
Reading	23.5	4.2	23.1–23.8	24.7	12.2	17.9–31.5	0.7093
Dictation	11.0	2.5	10.8–11.2	9.1	2.7	7.6–10.6	**0.0168**
Mathematics	16.5	2.6	16.3–16.7	13.5	3.6	7.4–11.6	**0.0061**

Data are means, standard deviations (SD), and 95% confidence intervals (95% CI). T-score means are compared using Student *t*-tests. Two children could not be assessed.

## Data Availability

Data will be shared on reasonable request to Raphaëlle Sarton (raphaelle.sarton@chu-reunion.fr) or Patrick Gérardin (patrick.gerardin@chu-reunion.fr).

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
