# Peer review of "Perinatal Mother-to-Child Chikungunya Virus Infection: Screening of Cognitive and Learning Difficulties in a Follow-Up Study of the Chimere Cohort on Reunion Island"

_viruses, 2025, doi:10.3390/v17050704_

Round 1

Reviewer 1 Report

Comments and Suggestions for Authors

This study describes a case series of CHIKV perinatally infected children assessed at 10 years of age.  This is a unique study population nested in the CHIMERE cohort that was developed during the CHIKV outbreak at Réunion Island between 2005-6.   Although biased by incomplete follow-up of all CHIMERE cohort infants, the neurodevelopmental abnormalities documented with the Evaluation of Cognitive Functions and Learning in Children) test (EDA) appears to confirm the long-term impact of maternal-neonatal CHIKV infection.  The authors describe well the limitations of this look-back study including the lack of EDA testing with relevant control infants from Réunion and MRI testing to correlate with EDA test results. 

The case series emphasizes the neurodevelopmental abnormalities observed in the EDA testing which may have be of less interest to the virology focused readership. 

Lines 194-206 -  These appear formatted as legend text for Figure 1 – but probably should be formatted as main text for Results 3.1 population characteristics. 

Comments on the Quality of English Language

Overall the manuscript would benefit from editing for language and style. 

Author Response

Review Report Form 1

Open Review

Quality of English Language

( ) The quality of English does not limit my understanding of the research.
(x) The English could be improved to more clearly express the research.

Comments and Suggestions for Authors

This study describes a case series of CHIKV perinatally infected children assessed at 10 years of age.  This is a unique study population nested in the CHIMERE cohort that was developed during the CHIKV outbreak at Réunion Island between 2005-6.   Although biased by incomplete follow-up of all CHIMERE cohort infants, the neurodevelopmental abnormalities documented with the Evaluation of Cognitive Functions and Learning in Children) test (EDA) appears to confirm the long-term impact of maternal-neonatal CHIKV infection.  The authors describe well the limitations of this look-back study including the lack of EDA testing with relevant control infants from Réunion and MRI testing to correlate with EDA test results. 

 The case series emphasizes the neurodevelopmental abnormalities observed in the EDA testing which may have be of less interest to the virology focused readership. 

 Lines 194-206 -  These appear formatted as legend text for Figure 1 – but probably should be formatted as main text for Results 3.1 population characteristics. 

The authors thank the reviewer for this remark. Effectively, the paragraph lines 195 to 295 refers to Results 3.1 Population characteristics and should appear after figure 1.

Comments on the Quality of English Language

Overall the manuscript would benefit from editing for language and style. 

Submission Date

15 November 2024

Date of this review

29 Nov 2024 18:20:16

Review Report Form 2

Comments and Suggestions for Authors

I would like to start by congratulating the authors for this exceptional manuscript, which addresses a highly relevant and understudied topic. The research on the cognitive and learning difficulties associated with perinatal chikungunya virus infection is not only original but also critical for advancing our understanding of the long-term impacts of arboviral infections in vulnerable populations. The methodology is well-structured, and the findings provide significant insights that could guide future research and public health interventions.

To further strengthen the manuscript and ensure clarity for a wider audience, I offer the following suggestions:

  1. Inclusion of Historical Context Regarding Severe and Fatal Cases (Lines 51–56):
    The manuscript mentions the recognition of rare, severe atypical presentations during the 2005–2006 CHIKV outbreak on Reunion Island. However, it is important to note that severe and fatal virologically confirmed cases of chikungunya had already been reported during outbreaks in India in the 1960s. These cases included neurological complications and haemorrhagic manifestations, as in the following studies:

-Chatterjee SN, Chakravarti SK, Mitra AC, Sarkar JK. Virological investigation of cases with neurological complications during the outbreak of haemorrhagic fever in Calcutta. J Indian Med Assoc. 1965 Sep 16;45(6):314-6. PMID: 5832545.

-Sarkar JK, Chatterjee SN, Chakravarty SK, Mitra AC. Virological studies in nine fatal cases of fever with haemorrhagic manifestation in Calcutta. Indian J Pathol Bacteriol. 1966 Apr;9(2):123-7. PMID: 5930492.

Adding this historical perspective could strengthen the discussion and provide a broader context for understanding the evolution of knowledge about severe chikungunya cases.                                                                                                                                                                                                    We are indebted of the reviewer for this historical perspective. To fulfil the recommendation without forgetting other Indian contributors, we added one of the proposed citations and quoted another from Vellore, Tamil Nadu, as Calcutta and Vellore were the two best described chikungunya epidemics in the sixties. Both cities observed neurological, hemorrhagic and fatal cases of virologically confirmed chikungunya, amidst concurrent dengue cocirculation.

  1. Clarification of the Socioeconomic Classification Used (Lines 120–121)
    The manuscript refers to "the best profession in the couple," ranked into four groups by the Institut national de statistiques et d’études économiques (Insee). While this classification might be familiar to French readers, it would be beneficial to provide a more detailed explanation of the criteria used for ranking these professions, ensuring that international readers can fully understand its implications for exposure assessment. 

We thank the reviewer for this important comment. Insee (Institut national des statistiques et études économiques) classification of the 7 French socio-professional categories was graded into 4 incremental groups according to average incomes in these groups. It was shown that exposure correlated with socio-economical deprivation and the most impoverished women were at higher risk of maternal-neonatal chikungunya when pregnant (Gérardin et al, PLoS Negl Trop Dis 2014). In this study, we preferred relying on the couple because a non-working woman may be due to her husband or partner’s supply.

  1. Calibration of the EDA in the Cited Study (Lines 136–140)
    The manuscript states that the EDA was developed and calibrated on a sample of 626 French healthy children, as referenced in [14]. However, upon reviewing this reference, I could not locate details regarding the calibration process. If this information is not explicitly provided in [14], it would be helpful for the authors to either clarify this point or provide an alternative citation that details the development and calibration of the EDA tool.

The reference was wrong. The paper to be mentioned was reference #15 now #17. Willig, T. N.; Billard, C.; Blanc, J. P.; Langue, J.; Touzin, M. Un nouvel outil d’évaluation des fonctions cognitives et des apprentissages pour le pédiatre : de la théorie à la pratique. Pediatre, 2013, 257, 2013-14.

Note that the term “calibration” has to be understood from the psychometric perspective and not from clinical research or epidemiologic perspective.

I am confident that addressing these points will further enhance the clarity and impact of this important work, and I look forward to seeing the final version published.

Haut du formulaire

Submission Date

15 November 2024

Date of this review

24 Dec 2024 21:40:49

Bas du formulaire

© 1996-2024 MDPI (Basel, Switzerland) unless otherwise stated

Reviewer 2 Report

Comments and Suggestions for Authors

I would like to start by congratulating the authors for this exceptional manuscript, which addresses a highly relevant and understudied topic. The research on the cognitive and learning difficulties associated with perinatal chikungunya virus infection is not only original but also critical for advancing our understanding of the long-term impacts of arboviral infections in vulnerable populations. The methodology is well-structured, and the findings provide significant insights that could guide future research and public health interventions.

To further strengthen the manuscript and ensure clarity for a wider audience, I offer the following suggestions:

1. Inclusion of Historical Context Regarding Severe and Fatal Cases (Lines 51–56):
The manuscript mentions the recognition of rare, severe atypical presentations during the 2005–2006 CHIKV outbreak on Reunion Island. However, it is important to note that severe and fatal virologically confirmed cases of chikungunya had already been reported during outbreaks in India in the 1960s. These cases included neurological complications and haemorrhagic manifestations, as in the following studies:

-Chatterjee SN, Chakravarti SK, Mitra AC, Sarkar JK. Virological investigation of cases with neurological complications during the outbreak of haemorrhagic fever in Calcutta. J Indian Med Assoc. 1965 Sep 16;45(6):314-6. PMID: 5832545.

-Sarkar JK, Chatterjee SN, Chakravarty SK, Mitra AC. Virological studies in nine fatal cases of fever with haemorrhagic manifestation in Calcutta. Indian J Pathol Bacteriol. 1966 Apr;9(2):123-7. PMID: 5930492.

Adding this historical perspective could strengthen the discussion and provide a broader context for understanding the evolution of knowledge about severe chikungunya cases.

2. Clarification of the Socioeconomic Classification Used (Lines 120–121)
The manuscript refers to "the best profession in the couple," ranked into four groups by the Institut national de statistiques et d’études économiques (Insee). While this classification might be familiar to French readers, it would be beneficial to provide a more detailed explanation of the criteria used for ranking these professions, ensuring that international readers can fully understand its implications for exposure assessment. 

3. Calibration of the EDA in the Cited Study (Lines 136–140)
The manuscript states that the EDA was developed and calibrated on a sample of 626 French healthy children, as referenced in [14]. However, upon reviewing this reference, I could not locate details regarding the calibration process. If this information is not explicitly provided in [14], it would be helpful for the authors to either clarify this point or provide an alternative citation that details the development and calibration of the EDA tool.

I am confident that addressing these points will further enhance the clarity and impact of this important work, and I look forward to seeing the final version published.

Author Response

(The authors gave the same response as above.)

Round 2

Reviewer 1 Report

Comments and Suggestions for Authors

The authors have responded to the reviewer comments with appropriate edits.